# Association between Cerebral Infarction Risk and Medication Adherence in Atrial Fibrillation Patients Taking Direct Oral Anticoagulants

**DOI:** 10.3390/healthcare9101313

**Published:** 2021-10-01

**Authors:** Yuuki Akagi, Akiko Iketaki, Reiko Nakamura, Shigeo Yamamura, Minori Endo, Keisuke Morikawa, Shun Oikawa, Takahiro Ohta, Shimpei Tatsumi, Takafumi Suzuki, Atsuhiro Mizushima, Keiichi Koido, Tatsuya Takahashi

**Affiliations:** 1Department of Pharmacy, National Hospital Organization Yokohama Medical Center, Kanagawa 245-8575, Japan; aki-aki-52126@msc.biglobe.ne.jp (A.I.); rik015.n@gmail.com (R.N.); dora2.oranger@gmail.com (K.K.); 2Faculty of Pharmaceutical Sciences, Josai International University, Chiba 283-8555, Japan; s_yama@jiu.ac.jp; 3Department of Pharmacy, National Hospital Organization Sagamihara Hospital, Kanagawa 252-0392, Japan; endo.minori.ek@mail.hosp.go.jp (M.E.); morikawa.keisuke.nm@mail.hosp.go.jp (K.M.); 4Department of Pharmacy, National Hospital Organization Mito Medical Center, Ibaraki 311-3193, Japan; s.oikawa0902@gmail.com; 5Department of Pharmacy, National Hospital Organization Tokyo Medical Center, Tokyo 152-8902, Japan; taka.ohta619@gmail.com; 6Department of Pharmacy, National Hospital Organization Tochigi Medical Center, Tochigi 320-8580, Japan; statsu8931@gmail.com; 7Department of Pharmacy, National Hospital Organization Utsunomiya Hospital, Tochigi 329-1193, Japan; suzuki.takafumi.rw@mail.hosp.go.jp; 8Department of Pharmacy, National Hospital Organization Shinshu Ueda Medical Center, Nagano 386-8610, Japan; mizushima.atsuhiro.nw@mail.hosp.go.jp; 9Department of Neurology, National Hospital Organization Yokohama Medical Center, Kanagawa 245-8575, Japan; tkhsh@yokohama-cu.ac.jp

**Keywords:** direct oral anticoagulant, adherence, cerebral infarction, nested case–control study, interview, pharmacist

## Abstract

Direct oral anticoagulants (DOACs) are available for nonvalvular atrial fibrillation patients. The advantage of DOACs is that regular anticoagulation monitoring is not required. However, adherence to the recommended regimen is essential. We investigated the association between medication adherence and the risk of cerebral infarction in patients taking DOACs. Patients admitted to any of the participating hospitals for cerebral infarction from September 2018 to February 2020 and prescribed DOACs before admission were defined as the case group, and patients hospitalized for diseases other than cerebral infarction, except for bleeding disorders, and prescribed DOACs before admission were defined as the control group. A nested case–control study was adapted, and 58 and 232 patients were included in the case and control groups, respectively. Medication adherence was assessed by the pharmacists through standardized interviewing. The adjusted odds ratio for the risk of cerebral infarction for low-adherence patients (<80% adherence rate) against good-adherence patients (100% adherence rate) was 9.69 (95% confidence interval, 3.86–24.3; *p* < 0.001). The patients’ age and other background characteristics were not found to be risk factors for cerebral infarction. In conclusion, low adherence is a risk factor for cerebral infarction in patients taking DOACs. Pharmacists should focus on maintaining ≥80% adherence to DOAC therapy to prevent cerebral infarction.

## 1. Introduction

Non-valvular atrial fibrillation (NVAF) is a common cardiac arrhythmia in the elderly, and it is known to be a significant risk factor for cerebral infarction [1]. Cardiogenic ischemic stroke is often severe, and the rate of recurrence is higher than that of other stroke types [2]. For the past 50 years and until recently, the vitamin K antagonist warfarin was the only agent available for preventing cerebral infarction in patients with NVAF, despite the need for regular anticoagulation monitoring and other considerations [3]. Direct oral anticoagulants (DOACs), which directly inhibit a single coagulation factor, were recently developed and have been available since around 2011. DOACs have many advantages compared to warfarin, including rapid onset of action, shorter half-life, no requirement for regular anticoagulation monitoring, a lower risk of intracranial hemorrhage, and fewer food and drug interactions [4]. Four DOACs (dabigatran, rivaroxaban, apixaban, and edoxaban) are currently approved in Japan.

Because regular anticoagulation monitoring is not required with DOAC therapy, the effect of DOACs is strongly dependent on the patient adherence to medication [5]. Persistent adherence is important and essential for the clinical efficacy of DOACs. The discontinuation rates of DOACs in phase III randomized controlled trials were reported to be high as 18–35% [5]. With inappropriate anticoagulant therapy, such as due to low adherence and underdosing, the risk of cerebral infarction increases. In some patients taking very low-dose dabigatran (75 mg twice daily), the development of thromboembolic events was observed [6]. A previous study demonstrated that inadequate control of the anticoagulant effect of warfarin leads to markedly higher rates of stroke and mortality [7]. Some studies that have used healthcare databases reported that lower adherence to DOAC therapy is associated with higher risk of stroke [8,9]. In these reports, the proportion of days covered (PDC) was used as the indicator of patient adherence to medication. Because PDC is defined as the number of doses dispensed in relation to the dispensing period, it might strictly differ from actual clinical practice. This means that the magnitude of the risk of poor adherence on cerebral infarction in clinical practice has not yet been clarified. The assessment and promotion of patient adherence to DOACs is therefore an essential role of pharmacists.

The main objective of this study was to reveal the risk of low adherence on cerebral infarction in patients with DOACs. In addition, the influence of overdose/underdose on cerebral infarction were also analyzed in patients with DOACs. To confirm that the patients actually adhered to the prescribed dose and frequency in DOAC therapy, data on medication adherence were collected through a standardized initial interview on admission by the pharmacists. Finally, we discussed the potential of pharmacists to reduce the risk of cerebral infarction by increasing patient adherence through interviews.

## 2. Materials and Methods

### 2.1. Study Design

Seven hospitals belonging to the National Hospital Organization participated in this study (Yokohama Medical Center, Sagamihara Hospital, Mito Medical Center, Tokyo Medical Center, Tochigi Medical Center, Utsunomiya Hospital, and Shinshu Ueda Medical Center). This multicenter, nested case–control study was performed after approval from the institutional review board (IRB) of each hospital (approval no. 30-6, Yokohama Medical Center; the representative hospital in our research team). This study followed the ethical guidelines outlined in the Declaration of Helsinki. The flowchart of patient inclusion and exclusion criteria is shown in Figure 1. Patients hospitalized for cerebral infarction from September 2018 to February 2020 (after approval from each hospital’s IRB) and prescribed a DOAC before admission were collected as the cerebral infarction case group. Patients hospitalized except for cerebral infarction who were prescribed a DOAC before admission were collected as the control group for 2–4 months, according to the approval date of each hospital (see Additional file/Appendix A). The prescription of DOACs is based on The Japanese Circulation Society and Japanese Heart Rhythm Society 2020 guideline on pharmacotherapy of cardiac arrhythmias. We excluded patients who were <40 years of age, were hospitalized for bleeding disorders, did not suffer from NVAF, and could not be interviewed for adherence because of communication disorders.

Case–control matching (1:4) was applied according to the three significant clinical characteristics of age, CHADS_2_ score, and history of cancer (Table 1). We used a propensity score method to adjust for these unbalanced variables. After the calculation of the propensity score using the logistic regression method, 4 controls with similar propensity were selected for a case. The caliper width was the 20% standard deviation of the logit of the propensity score.

### 2.2. Assessment of Adherence to DOAC

Before commencement of the study, we standardized the adherence status among the participating hospitals as follows: (I) I have never forgotten, (II) I have forgotten only 1 day in a week, or (III) I have forgotten ≥2 days in a week (including self-interrupted). A score of (I) indicates 100% adherence rate, (II) indicates ≥80% but <100%, and (III) indicates <80%. Zero % adherence rate means taking DOAC is self-interrupted. The patients were interviewed on admission about their adherence of DOAC. An assessment of medication adherence was carried out as described above by the pharmacists, which involved asking patients about their medications for the 3 weeks prior to hospital admission. The assessment of adherence was conducted on DOAC only, not all their medications.

### 2.3. Assessment of DOAC Dosage

Appropriate standard doses and low doses of the DOACs were defined as administration according to the package insert of each DOAC. Overdosing (off-label standard dose) was defined as the administration of a standard dose of the DOAC, although the low dosage criteria were met. Underdosing (off-label low dose) was defined as the administration of a low dose of the DOAC, although the standard dosage criteria were met conversely [10]. Assessment of the DOAC dosage was recorded as (a), (b), or (c) as follows: (a) overdose, (b) appropriate dose (including appropriate standard dose and low dose), or (c) underdose (Table 2).

### 2.4. Data Collection

The following data of the subjects were collected from the electronic medical records of each hospital: age, sex, body weight, serum creatinine level, the name and dosage of the DOAC prescribed before admission, diseases that led to hospitalization (cerebral infarction, bleeding, or others), living alone (absence of a medication supporter), incidence of complications (congestive heart failure, hypertension, diabetes mellitus, and stroke/transient ischemic attack for CHADS_2_ scores) [11], combination of antiplatelet drugs (low-dose aspirin, ticlopidine, clopidogrel, prasugrel, and cilostazol), agents which might reduce the anticoagulant effect of DOAC (rifampicin, carbamazepine, phenobarbital, and phenytoin; described in the package insert in Japan) [12,13], and history of cancer. The prescription information about the one dose package service for dispensing (ODP) and the number of daily doses (all the internal medicines, not only DOAC) were also collected.

### 2.5. Statistical Analyses

The Chi-square test for categorical variables and Student’s t-test for numerical variables were used to compare the clinical characteristics between the patients hospitalized for cerebral infarction and the other patients. The adjusted odds ratio (aOR) and 95% confidence interval (CI) were calculated using a multivariable logistic regression analysis. Statistical analyses were performed using JMP Pro 14.2.0 (SAS Institute Japan Ltd., Tokyo, Japan). Matching with the propensity score was carried out using EZR [14]. Differences between the groups were assessed using two-sided tests with an alpha level of 0.05.

## 3. Results

### 3.1. Patients’ Characteristics

During the study period, the data of 618 patients were collected. After excluding patients according to our protocol, 86 patients were included in the cerebral infarction cases, and 380 patients were included as controls (Figure 1). The clinical characteristics of patients are described in Table 1. Only one patient was taking agents which might have reduced the anticoagulant effects of DOACs (carbamazepine, in the controls). In total, 58 cerebral infarction matched cases and 232 matched controls were selected by applying the propensity score (Table 3). Rivaroxaban was the most prescribed DOAC in our study. However, we observed little association with the cerebral infarction for which the DOACs were prescribed.

### 3.2. Medication Adherence and other Patients’ Background on the Risk of Cerebral Infarction

In the cerebral infarction matched cases, the number of patients with an adherence status of (I), (II), and (III) was 35 (60.3%), 7 (12.1%), and 16 (27.6%), respectively. The number of patients in the control group with an adherence status of (I), (II), and (III) was 195 (84.1%), 25 (10.8%), and 12 (5.2%), respectively. Eight patients had self-interrupted the medication: five in the cerebral infarction matched cases and three in the matched control. The results of the multivariable logistic regression analysis are shown in Table 4. The aOR of medication adherence status (II: ≥80% but <100% adherence rate)/(I: 100% adherence rate) was 1.69 (95% CI, 0.62–4.60; *p* = 0.301), and (III: <80% adherence rate)/(I) was 9.69 (95% CI, 3.86–24.3; *p* < 0.001), respectively. In the cerebral infarction matched cases, lower adherence significantly increased the risk of hospitalization for cerebral infarction. Other factors (DOAC dosage, ODP, living status, the number of daily doses, age, etc.) did not influence hospitalization for cerebral infarction between the matched case and control groups.

## 4. Discussion

This nested case–control study found that several patients missed a DOAC dose at least once a week, similar to the results found by previous reports. We found that some patients had often forgotten or discontinued the medication on their own (self-interrupted). Furthermore, the lower the adherence to medication, especially at a <80% adherence rate, the higher the risk of cerebral infarction. Off-label underdosing of DOAC therapy is a clinical concern. However, an increased risk of cerebral infarction was not observed in our results.

Non-adherence to medication is common in patients with various chronic diseases [15]. Discontinuation of medication appears to be one of the largest factors determining the patient prognosis, as adherence to anticoagulant therapy is crucial for stroke prevention. Previous investigations have described the proportion of non-adherent patients. In particular, PDC < 80%, was observed in 27.6–27.8% of the patients [8,16] in the Veterans’ Health Administration data. It has been suggested that continuous good adherence (PDC ≥ 80%) to oral anticoagulation is associated with a reduced stroke risk of approximately 40% in patients newly diagnosed with NVAF [9]. In our study, the proportion of patients who missed the medication doses (adherence rate < 100%; adherence status II and III) was 21.7% (60/290). The proportion of those with a DOAC adherence rate < 80% (adherence status III) was 9.7% (28/290). More patients with low adherence were observed in the cerebral infarction matched case. The persistent use of preventive drugs, such as anticoagulants, declines rapidly [17,18,19], because the perceived benefits of anticoagulant therapy might not be well understood by the patients. DOACs do not require routine anticoagulation monitoring, which may promote low adherence among patients [20]. Adverse events, including gastrointestinal symptoms and bleeding, were the most common reasons for discontinuing DOACs [21]. It is assumed that patients, especially those hospitalized for cerebral infarction, have experienced these factors. Life expectancy is increasing in Japan, and the number of patients prescribed an anticoagulant might increase in the future. It is clear that improvement of adherence to DOAC therapy by each patient is important [9]. Although doctors and pharmacists may think or believe that the medications are usually taken as prescribed, this is not always the case. Doctors and pharmacists must therefore accept that adherence tends to decline over time [5], leading to an increased risk of cerebral infarction. 

Good adherence has been defined as a PDC or medication possession ratio (MPR) ≥80% in many studies. However, for DOAC therapy, this rate is not based on evidence [15]. Taking warfarin with ≥20% missed bottle openings was associated with anticoagulation [22]. The risk of cerebral infarction was found to be higher in patients with poor adherence (adherence status III) than in those with good or moderate adherence (adherence status I or II). There was no difference in the risk of cerebral infarction between the matched cases and controls in patients with good adherence (adherence status I) and moderate adherence (adherence status II). These results suggest that a medication adherence rate of 100% is desirable; however, a rate of ≥80% in DOAC therapy is acceptable for reducing the risk of cerebral infarction. In other words, the risk of cerebral infarction was increased when the adherence rate was lower than 80%. As a result, our claim can be supported by the previous studies using PDC as the indicator of adherence [16,23]. Because the half-life of DOACs is shorter than that of warfarin, the anticoagulant effect of DOAC rapidly decreases after the concentration peaks. If a DOAC dose is missed, then the duration of an inadequate anticoagulant effect is longer. Hence, the risk for cerebral infarction is increased in NVAF patients with low adherence, and a higher rate of adherence is desirable in patients with moderate adherence (adherence status II).

As mentioned earlier, the patients were interviewed on admission by the pharmacists about their adherence to DOAC. The concept of medication adherence includes many factors, such as understanding the significance of medication and aggressive treatment, as well as compliance with medication. However, in this study, our emphasis is on whether the DOAC doses were actually taken as prescribed. Adherence to DOAC has been evaluated using the proportion of prescription days in various medical databases in many studies; however, in those studies, the researchers assumed that the patients were completely compliant if a prescription was given. In addition, the PDC would theoretically be almost 100% in the reservation-only hospital if a patient visited regularly. There are various methods for measuring adherence, such as PDC, MPR, medication event monitoring systems, and self-reporting [24]. To determine whether the DOAC doses were actually taken, we collected patient self-report data. Nevertheless, this method is limited by the accuracy of adherence assessment because the patients might not assess themselves accurately or may provide false data. However, the patients might be more willing to admit to missing doses to a pharmacist rather than a doctor. Hence, it is very important to evaluate medication adherence through the pharmacists.

It is assumed that numerous patient and healthcare factors are associated with medication adherence. A previous report described that younger age, male sex, low overall stroke risk, poor cognitive function, poverty, homelessness, higher education, employment, and reluctance to receptivity of medical information were associated with poor adherence to warfarin therapy [25]. It appears here that the reluctant receptivity of medical information is the underlying cause, and adherence tends to decline if a patient is busy, poor, or has psychosocial determinants. In this study, we matched the backgrounds of the patients in the case and control groups and adjusted for the major confounding factors. However, no risk factors were found for cerebral infarction except for adherence. Association with low adherence of age and other risks of cerebral infarction could not be observed because case–control matching was applied with those considerations. Adverse events, even if these were not severe, and patient preference were the important causes of poor adherence [21]. This means that the understanding and patient education of anticoagulant therapy is insufficient. Physicians and pharmacists should consider the possibility of low adherence and ask patients about their adherence regularly, as this is the most realistic way to confirm the anticoagulant effect of DOAC therapy. This was one of the possible reasons why only in the case of ODP, having medication support and once-daily doses did not significantly decrease the risk of cerebral infarction. A previous report described that the assessment of appropriate DOAC dosage prescriptions and ongoing monitoring by pharmacists (pharmacist-led DOAC service) increased the rate of appropriate DOAC dosing and improved patient adherence to medication [26]. A previous study showed that specific pharmacist-led education about adverse events and adherence monitoring were associated with greater patient adherence to dabigatran [27]. Regular reviews by healthcare providers, including pharmacists and nurses, might also play an important role in monitoring patient adherence to DOACs [28]. It seems that measures such as patient education are necessary to maintain ≥80% adherence.

One of the most troublesome cases is where there is self-interruption of medication. In some cases, the medicine was stored and not consumed, although the patient visited the doctor, and the medication was prescribed. In other cases, the patient stopped visiting the doctor. The former scenario appears to be rare, but can happen with some patients, especially those who do not need to bear the cost of medical expenses. The latter scenario is more common because the patient has not visited the doctor. In this case, it is difficult for medical staff to find the patient, and there is no follow-up of the patient. When the medication history is questioned (medical checkup, hospitalization, or visit to a clinic or a pharmacy), the physicians and the pharmacists might find that an anticoagulant was prescribed in the past. Thus, some cases of self-interruption of anticoagulant therapy might be identified. As already mentioned, five patients in the cerebral infarction matched cases and three patients in the matched control had self-interrupted the medication. This indicates that if the patients who self-interrupted the anticoagulant are lost to follow-up, it will be difficult for the medical staff to locate them until the onset of cerebral infarction. The three patients in the control group who had self-interrupted the medication were informed about the importance of medication adherence by the pharmacist.

There are some limitations to this study. First, this was a nested case–control study by design, and any influence of unknown and unpredictable background factors cannot be excluded. Because the enrolled study patients were those who had already been taking DOACs, the proportion of patients with high bleeding risk in this study may be lower than that in real-world settings. Furthermore, because the control group was selected from the inpatients by considering their feasibility, it is possible that unmatched characteristics of these patients might differ from those of the average patient population. Second, the reliability and accuracy in the assessment of medication adherence, as already mentioned, are important considerations. Measurement using a reliable and validated tool is desirable, however the communication with cerebral infarction patients is often difficult and it was considered hard to answer many questions, so we evaluated a single question of medication adherence, that is self-reporting. Third, the sample size of the study was relatively small compared to studies that have used real-world databases. Adequate assessment of the factors associated with non-adherence could not be performed, though this might not have affected the main outcomes of the study.

## 5. Conclusions

There was a significant association between lower adherence and the risk of cerebral infarction in our investigation with adjustments for possible confounding factors and matching the backgrounds of patients in the case and control groups. The results of this study indicate that maintaining ≥80% adherence to DOAC therapy is important for the prevention of cerebral infarction. Ensuring that patients take oral anticoagulants regularly without repeatedly missing the doses is significant and necessary not only for physicians and pharmacists, but also for cohabitants and supporters.

## Figures and Tables

**Figure 1 healthcare-09-01313-f001:**
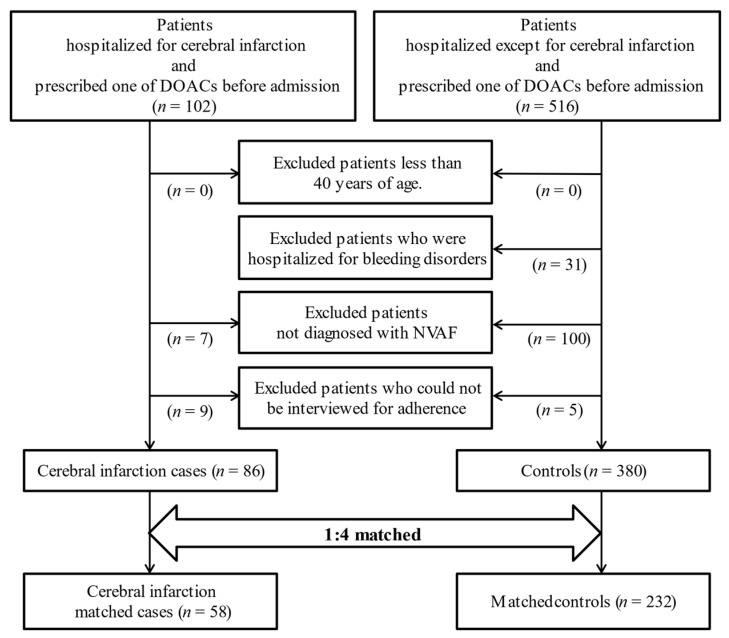
Flowchart of this study. DOAC, direct oral anticoagulant; NVAF, non-valvular atrial fibrillation.

**Table 1 healthcare-09-01313-t001:** Clinical characteristics of the study patients (before matching).

Characteristics	Cerebral Infarction Cases (before Matching)	Controls (before Matching)	*p* Value
	(*N* = 86, Mean ± S.D.)	(*N* = 380, Mean ± S.D.)	
Age	81.4 ± 6.6	77.6 ± 9.5	<0.001 **
Sex M/F	51/35	250/130	0.256 *
Creatinine clearance (CCr, mL/min)	53.0 ± 20.7	55.3 ± 26.3	0.471 **
DOAC			
Dabigatran	10	42	0.519 *
Rivaroxaban	35	125	
Apixaban	21	103	
Edoxaban	20	110	
CHADS_2_ score	2.90 ± 1.15	2.53 ± 1.30	0.018 **
With antiplatelets	11 (12.8%)	61 (16.0%)	0.450 *
History of cancer	11 (12.8%)	108 (28.4%)	0.003 *

Mean ± S.D. or number (%); *, Chi-square test; **, Student’s *t*-test.

**Table 2 healthcare-09-01313-t002:** Assessment of the DOAC dosage.

DOAC	Overdose	Appropriate Dose		Underdose
		Appropriate Standard Dose	Appropriate Low Dose	
Dose/Criteria	Standard/Low	Standard/Standard	Low/Low	Low/Standard
Dabigatran	150 mg twice a day	150 mg twice a day	110 mg twice a day	110 mg twice a day
	Age ≥ 70	Does not meet the low-dose criteria	Age ≥ 70	Does not meet the low-dose criteria
	30 ≤ CCr ≤ 50 mL/min		30 ≤ CCr ≤ 50 mL/min	
Rivaroxaban	15 mg once a day	15 mg once a day	10 mg once a day	10 mg once a day
	15 ≤ CCr ≤ 50 mL/min	Does not meet the low-dose criteria	15 ≤ CCr ≤ 50 mL/min	Does not meet the low-dose criteria
Apixaban	5 mg twice a day	5 mg twice a day	2.5 mg twice a day	2.5 mg twice a day
	Age ≥ 80	Does not meet the low-dose criteria	Age ≥ 80	Does not meet the low-dose criteria
	BW ≤ 60 kg		BW ≤ 60 kg	
	SCr ≥ 1.5 mg/dL		SCr ≥ 1.5 mg/dL	
Edoxaban	60 mg once a day	60 mg once a day	30 mg once a day	30 mg once a day
	BW ≤ 60 kg	Does not meet the low-dose criteria	BW ≤ 60 kg	Does not meet the low-dose criteria
	15 ≤ CCr ≤ 50 mL/min		15 ≤ CCr ≤ 50 mL/min	

CCr, creatinine clearance; BW, body weight; SCr, serum creatinine.

**Table 3 healthcare-09-01313-t003:** Clinical characteristics of the study patients (after matching).

Characteristics	Cerebral Infarction Matched Cases	Matched Controls	*p* Value
	(*N* = 58, Mean ± S.D.)	(*N* = 232, Mean ± S.D.)	
Age	78.7 ± 6.5	78.0 ± 10.0	0.206 **
Over 80 years old	30	120	1.000 *
Sex M/F	37/21	143/89	0.762 *
Creatinine clearance (CCr, mL/min)	53.2 ± 20.3	55.7 ± 26.3	0.703 **
DOAC			
Dabigatran	8	22	0.271 *
Rivaroxaban	26	88	
Apixaban	11	63	
Edoxaban	13	66	
CHADS_2_ score	2.60 ± 1.12	2.60 ± 1.11	1.000 **
With antiplatelets	6 (10.3%)	43 (18.5%)	0.137 *
History of cancer	6 (10.3%)	24 (10.3%)	1.000 *

Mean ± S.D. or number (%); *, Chi-square test; **, Student’s *t*-test.

**Table 4 healthcare-09-01313-t004:** Patients’ factors affecting hospitalization for cerebral infarction.

Factor	Adjusted Odds Ratio	*p* Value
	(95% CI)	
Medication adherence (II)/(I)	1.69 (0.62–4.60)	0.301
Medication adherence (III)/(I)	9.69 (3.86–24.3)	<0.001 *
Medication adherence (III)/(II)	5.72 (1.64–19.9)	0.006 *
DOAC dosage (b)/(a)	0.82 (0.25–2.74)	0.758
DOAC dosage (c)/(a)	0.46 (0.11–1.84)	0.271
DOAC dosage (c)/(b)	0.55 (0.23–1.32)	0.183
One dose package service for dispensing	0.70 (0.35–1.42)	0.330
Living alone (absence of a medication supporter)	0.93 (0.46–1.88)	0.845
Number of daily doses (twice or more)	1.41 (0.71–2.83)	0.330
Over 80 years old	0.91 (0.45–1.85)	0.794
CHADS_2_ score	1.03 (0.74–1.43)	0.850
History of cancer	1.32 (0.47–3.65)	0.597

CI, confidence interval; *, *p* < 0.05; a score of medication adherence (I) indicated a 100% adherence rate, (II) indicated ≥80% but <100%, and (III) indicated <80%. DOAC dosage was as recorded (a), (b), or (c) as follows: (a) overdose, (b) appropriate dose (including appropriate standard dose and low dose), or (c) underdose. No interactions were found between medication adherence—DOAC dosage, medication adherence—the number of daily doses, and DOAC dosage—the number of daily doses on the risk of cerebral infarction.

## Data Availability

The datasets used and analyzed in our study are available from the corresponding author on reasonable request.

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
