# Peer review of "Association between Cerebral Infarction Risk and Medication Adherence in Atrial Fibrillation Patients Taking Direct Oral Anticoagulants"

_healthcare, 2021, doi:10.3390/healthcare9101313_

Round 1

Reviewer 1 Report

Association between cerebral infarction risk and medication adherence in atrial fibrillation patients taking direct oral anticoagulants

This is a well designed study using nested case–control design, and 58 and 232 patients were included in the case and control groups, respectively.

I have some minor comments:

Method:

Assessment of adherence to DOAC

Before commencement of the study, we standardized the adherence status among  the participating hospitals as follows: (I) I have never forgotten, (II) I have forgotten only 1 day in a week, or (III) I have forgotten ≥2 days in a week (including self-interrupted). A  score of (I) indicates 100% adherence rate, (II) indicates ≥80% but <100%, and (III) indicates  <80%. An assessment of medication adherence was carried out by the pharmacists, which involved asking patients about their medications for the 3 weeks prior to hospital admission.

-is this a validated tool? Who developed this tool?

Pls state the name of the Guideline used to guide the prescribing of DOACs

The results of this study indicate that maintaining ≥80% adherence to DOAC therapy is important for the prevention of cerebral infarction.-pls state any other studies that can support this claim

Reviewer 2 Report

This was an interesting, multi-centre study investigating the relationship between medication adherence and hospitalisation for cerebral infarction in patients prescribed direct oral anticoagulants (DOACs).  It included a matched control group of patients prescribed DOACs, but hospitalised for any other reason.  The study was interesting and well-conducted, and a helpful addition to the literature.  I think it would be suitable to be published once my concerns below are addressed. 

  1. In the abstract, should clarify that the control group were also prescribed DOACs for NVAF, and that it was a matched control group. Currently, this is not clear, so the control group does not appear appropriate. 
  2. Why is the lack of requirement for monitoring cited as a cause of the association between effectiveness and adherence? (line 57)
  3. The description of the measure of adherence should be clearer. Were participants asked to select which of statements 1/2/3 were correct for the past 3 weeks overall (i.e. a multiple choice question with 3 response options), or asked this question 3 times, once for each of the past 3 weeks?  Were the questions asked about all their medications, or only the DOACs? 
  4. Where the overdose/underdose classifications based on whether the prescribing was appropriate, or whether patients were taking more/less medication than prescribed? There should be justification for why this variable was measured (i.e. including reference to it in the study aims), and how it was analysed.  The adherence variable is also not mentioned in the analysis section. 
  5. How was the matching conducted?
  6. Please give percentages within each group for each adherence score (lines 166/7)
  7. In section 3.2 it says: “In the cerebral infarction matched cases, lower adherence significantly increased the risk of hospitalization.” However, all patients in the study were hosptialised, so talking about risk of hospitalisation does not seem to make sense.  Do you mean: In the matched cases, lower adherence significantly increased the risk of hospitalisation for cerebral infarction?  Please clarify. 
  8. This study used a very rough measurement of adherence, using only a single question. Even if self-report is measured, potentially a more reliable and validated tool could be used.  There are different validation tools to measure adherence – was there a reason why these were not used?  This limitation should be discussed in the discussion. 
  9. Lines 270-285 discuss self-interruption. It was not clear in the methods how this was assessed.  It would have been helpful to state these figures in the results as descriptive data. 

Reviewer 3 Report

The question raised by the authors are of interest. The study design sound appropriate. Nevertheless I have some comments about the statistical analysis and methods

Why the authors did not use statistical tests for matched data to analyse and compare their groups? It would have been more appropriate. If there no reason about this, I think you should do this.

In the statistical model reported in table 4, I think there is a colinearity between the DOAC dosage and the number of daily doses... Why did the authors introduced these variables concomitantly?

Moreover did the authors test then interaction between the medication adherence and the dosage to test whether the inital dosage may affect the effect of adherence on the risk of cerebral infarction.

How did the auhors measure the DOAC dosage, through the pharmacist using the prescription?

Does the authors have the reason why the lack of adherence for some patients? Are there any medical reason? Is it due to side effects? I think this aspect if of importance to understand the reasons why as welle as factors associated with this lower adherence for some patients. I think the discussion should focus more on this aspect.
